# Fast Decomposable Submodular Function Minimization using Constrained Total Variation

**K S Sesh Kumar**
Data Science Institute
Imperial College London, UK
`s.karri@imperial.ac.uk`

**Francis Bach**
INRIA and Ecole normale superieure
PSL Research University, Paris France.
`francis.bach@inria.fr`

**Thomas Pock**
Institute of Computer Graphics and Vision,
Graz University of Technology, Graz, Austria.
`pock@icg.tugraz.at`

## Abstract

We consider the problem of minimizing the sum of submodular set functions assuming minimization oracles of each summand function. Most existing approaches reformulate the problem as the convex minimization of the sum of the corresponding Lovász extensions and the squared Euclidean norm, leading to algorithms requiring total variation oracles of the summand functions; without further assumptions, these more complex oracles require many calls to the simpler minimization oracles often available in practice. In this paper, we consider a modified convex problem requiring a constrained version of the total variation oracles that can be solved with significantly fewer calls to the simple minimization oracles. We support our claims by showing results on graph cuts for 2D and 3D graphs.

## 1  Introduction

A discrete function $F$ defined on a finite ground set $V$ of $n$ objects is said to be submodular if the marginal cost of each object reduces with the increase in size of the set it is conditioned on, i.e., $F : 2^V \to \mathbb{R}$ is submodular if and only if the marginal cost of an object $\{x\} \in V$ conditioned on the set $A \subseteq V \setminus \{x\}$ denoted by $F(\{x\}|A) = F(\{x\} \cup A) - F(A)$ reduces as the set $A$ becomes bigger. The diminishing returns property of submodular functions has been central to solving several machine learning problems such as document summarization [1], sensor placement [2] and graphcuts [3] (see [4] for more applications). Without loss of generality, we consider normalized submodular functions, i.e., $F(\varnothing) = 0$.

Submodular function minimization (SFM) can be solved exactly using polynomial algorithms but with high computational complexity. One of the standard algorithms is the Fujishighe-Wolfe algorithm [5, 6] but most recently, SFM has been tackled using cutting plane methods [7] and geometric scaling [8]. All the above algorithms rely on value function oracles, that is, access to $F(A)$ for arbitrary subsets $A$ of $V$, and solve SFM with high running-time complexities, e.g., $O(n^4 \log^{O(1)} n)$ and more. These algorithms are typically not trivial to implement and do not scale to problems with large ground sets (such as $n = 10^6$ in computer vision applications). For scalable practical solutions, it is imperative to exploit the structural properties of the function to minimize.

Submodularity is closed under addition [5, 4]. We make a structural assumption that is practically useful in many machine learning problems [9, 10] (e.g., when a 2D-grid graph is seen as the concatenation of vertical and horizontal chains) and consider the problem of minimizing a sum of

submodular functions [11], i.e.,

$$\min_{A \subset V} F(A) := \sum_{i=1}^{r} F_i(A), \tag{1}$$

assuming each summand $F_i$, $i = 1, \ldots, r$, is "simple", i.e., with available efficient oracles which are more complex than plain function evaluations. The simplest of these oracles is being able to minimize the submodular function $F_i$ plus some modular function, and we will consider these oracles in this paper. This is weaker than the usual "total variation" oracles detailed below.

One of the standard approaches to solve the discrete optimization problem in Eq. (1) is to consider an equivalent continuous optimization problem that minimizes the Lovász extension [12] $f$ of the submodular function over the $n$-dimensional unit hypercube (see a definition in Section 2). This approach uses a well known result in the submodularity literature that the minimizers of the set function $F$ can be directly obtained from the minimizers of its Lovász extension $f$; the continuous optimization problem is given by

$$\min_{w \in [0,1]^n} f(w) := \sum_{i=1}^{r} f_i(w), \tag{2}$$

where $f_i$ is the Lovász extensions of submodular function $F_i$, for each $i \in [r]$. Lovász extension of submodular functions are convex but non-smooth and piecewise linear functions. Therefore, we can use subgradients as they can be calculated using greedy algorithms in $O(n \log(n))$ time and $O(n)$ calls to the value function oracle per iteration [13]. However, this is slow with $O(1/\sqrt{t})$ convergence rate where $t$ is the number of iterations of the optimization algorithms. Moreover, in signal processing applications, high precision is needed, hence the need for faster algorithms.

An alternative approach is to consider a continuous optimization problem [4, Chapter 8] of the form

$$\min_{w \in \mathbb{R}^n} f(w) + \sum_{j=1}^{n} \psi(w_j), \tag{3}$$

where $\psi : \mathbb{R} \to \mathbb{R}$ is a convex function whose Fenchel-conjugate [14] is defined everywhere (in order to have a well-defined dual as later shown in Eq. (9)). This is equivalent to solving all the following discrete optimization problems parameterized by $\alpha \in \mathbb{R}$,

$$\min_{A \subset V} F(A) + |A|\psi'(\alpha). \tag{4}$$

Given the solutions $A^\alpha$ for all $\alpha \in \mathbb{R}$ in Eq. (4), then we may obtain the optimal solution $w^*$ of Eq. (3) using $w_j^* = \sup\{\alpha \in \mathbb{R}, j \in A^\alpha\}$. Conversely, given the optimal solution $w^*$ of Eq. (3), we may obtain the solutions $A^\alpha$ of the discrete optimizaton problems in Eq. (4) by thresholding at $\alpha$, i.e., $\{w^* \geq \alpha\}$. As a consequence of this we can obtain the solution of Eq. (1), when $\alpha$ is chosen so that $\psi'(\alpha) = 0$ (typically $\alpha = 0$ because $\psi$ is even). Note that this algorithmic scheme seems wasteful because we take a continuous solution of Eq. (3) and only keep the signs of its solution. One contribution of the paper is to propose a function $\psi$ that focuses only on values of $w$ close to zero.

**Our problem setting and approach.** We assume the availability of SFM oracles of the individual summand functions, which we refer to as SFM$\mathbf{D}_i$, for each $i \in [r]$ that gives the optimal solution of

$$\text{SFM}\mathbf{D}_i : \underset{A \subset V}{\arg\min}\, F_i(A) - u^\top 1_A, \tag{5}$$

where $u \in \mathbb{R}^n$ is any $n$-dimensional vector and $1_A \in \{0,1\}^n$ is the indicator function of the set $A$. Note that the complexity of the oracle does not typically depend on the vector $u$. We consider the following continuous optimization problem, which we refer to as SFM$\mathbf{C}_i$ for each $i \in [r]$,

$$\text{SFM}\mathbf{C}_i : \underset{w \in \mathbb{R}^n}{\arg\min}\, f_i(w) - t^\top w + \sum_{j=1}^{n} \psi(w_j), \tag{6}$$

where $t \in \mathbb{R}^n$ is any $n$-dimensional vector. In our setting, we consider $\psi$ as the following convex function,

$$\psi(w) = \begin{cases} \frac{1}{2}w^2 & \text{if } |w| \leqslant \varepsilon, \\ +\infty & \text{otherwise}, \end{cases} \tag{7}$$

where $\varepsilon \in \mathbb{R}_+$. In Section 3.1, we show that the continuous optimization problem $\mathrm{SFM}\mathbf{C}_i$ can be optimized using discrete oracles $\mathrm{SFM}\mathbf{D}_i$ using a modified divide-and-conquer algorithm [15]. In Section 3.2, we use various optimization algorithms that use $\mathrm{SFM}\mathbf{C}_i$ as the inner loop to solve the continuous optimization problem in Eq. (3) consequently solving the SFM problem in Eq. (1).

**Related work.** Most of the earlier works have considered quadratic functions for the choice of $\psi$ [15, 16, 17, 18, 19, 20], i.e., $\psi(v) = \frac{1}{2}v^2$. As a result, $\mathrm{SFM}\mathbf{C}_i$ in Eq. (6) is referred to as total variation or TV oracle as they solve the problems of the form $\min_{w \in \mathbb{R}^n} f(w) + \frac{1}{2}(t-w)^2$ [21]. These oracles are efficient for cut functions defined on chain graphs [22, 23] with $O(n)$ complexity. However, this does not hold for general submodular functions. One way to solve continuous optimization problems like $\mathrm{SFM}\mathbf{C}_i$ is to use a sequence of at most $n$ discrete minimization oracles like $\mathrm{SFM}\mathbf{D}_i$ through divide-and-conquer algorithms (see Section 3.1).

Recent work has also focused on using directly discrete minimization oracles of the form $\mathrm{SFM}\mathbf{D}_i$, such as [16] that considers a total variation problem with active set methods; [24] used discrete minimization oracles $\mathrm{SFM}\mathbf{D}_i$ but solved a different convex optimization problem. [25] also considers the total variation problem but uses incidence relations and oblique projections for quicker convergence. [26] reduces the search space for the SFM problem, i.e., $V$, using heuristics. Our choice of $\psi$ results in a similar reduction of the search space that results in a more efficient solution.

**Contributions.** Our main contribution is to propose a new convex optimization problem that can be used to find the minimum of a sum of submodular set-functions. For graph cuts, this new problem can be seen as a constrained total variation problem that is more efficient that the regular total variation (lesser number of discrete minimization oracle calls). This is beneficial when minimizing the sum of constrained total variation problems, and consequently beneficial for the corresponding discrete minimization problem, i.e., minimizing the sum of submodular functions. For the case of sum of two functions, we show that recent acceleration techniques from [27] can be highly beneficial in our case. This is validated using experiments on segmentation of two dimensional images and three dimensional volumetric surfaces.

Note that we use cuts mainly due to easy access to minimization oracles of cut functions [28], but our result applies to all submodular functions.

## 2  Review of Submodular Function Minimization (SFM)

In this section, we review the relevant concepts from submodular analysis (for more details, see [4, 5]). All possible subsets of the ground set $V$ can be considered as the vertices $\{0, 1\}^n$ of the hypercube in $n$ dimensions (going from $A \subseteq V$ to $1_A \in \{0, 1\}^n$). Thus, any set-function may be seen as a function $F$ defined on the vertices of the hypercube $\{0, 1\}^n$. It turns out that $F$ may be extended to the full hypercube $[0, 1]^n$ by piecewise-linear interpolation, and then to the whole vector space $\mathbb{R}^n$ [4].

This extension $f$ is piecewise linear for any set-function $F$. It turns out that it is convex if and only if $F$ is submodular [12]. Any piecewise linear convex function may be represented as the support function of a certain polytope $K$, i.e., as $f(w) = \max_{s \in K} w^\top s$ [14]. For the Lovász extension of a submodular function, there is an explicit description of $K$, which we now review.

**Base polytope.** We define the *base polytope* as $B(F) = \big\{ s \in \mathbb{R}^n, \ s(V) = F(V), \ \forall A \subset V, s(A) \leqslant F(A) \big\}$, where we use the classical notation $s(A) = s^\top 1_A$. A key result in submodular analysis is that the Lovász extension is the support function of $B(F)$, that is, for any $w \in \mathbb{R}^n$,

$$f(w) = \sup_{s \in B(F)} w^\top s. \tag{8}$$

The maximizers above may be computed in closed form from an ordered level-set representation of $w$ using a "greedy algorithm", which (a) first sorts the elements of $w$ in decreasing order such that $w_{\sigma(1)} \geq \ldots \geq w_{\sigma(n)}$ where $\sigma$ represents the order of the elements in $V$; and (b) computes $s_{\sigma(k)} = F(\{\sigma(1), \ldots, \sigma(k)\}) - F(\{\sigma(1), \ldots, \sigma(k-1)\})$. This leads to a closed-form formula for $f(w)$ and a subgradient.

**SFM as a convex optimization problem.** A key result from submodular analysis [12] is the equivalence between the SFM problem $\min_{A \subseteq V} F(A)$ and the convex optimization problem $\min_{w \in [0,1]^n} f(w)$. One can then obtain an optimal $A$ from level sets of an optimal $w$. Moreover, this leads to the dual problem $\max_{s \in B(F)} \sum_{i=1}^{n} (s_i)_-$. Note that for our algorithm to work, we need oracles SFM$\mathbf{D}_i$ that output both the primal variable ($A$ or $w$) and the dual variable $s \in B(F)$.

**Convex optimization and its dual.** We consider the continuous optimization problem in Eq. (3). Its dual problem derived using Eq. (8) is given by

$$\max_{s \in B(F)} - \sum_{j=1}^{n} \psi^*(-s_j). \tag{9}$$

In this paper, we consider the convex function $\psi : \mathbb{R} \to \mathbb{R}$ defined in Eq. (7). Its Fenchel-conjugate $\psi^*$ is given by

$$\psi^*(s) = \begin{cases} \frac{1}{2}s^2 & \text{if } |s| \leqslant \varepsilon, \\ \varepsilon|s| - \frac{\varepsilon^2}{2} & \text{otherwise.} \end{cases} \tag{10}$$

# 3 Fast Submodular Function Minimization with Constrained Total Variation

In this section, we propose an algorithm to optimize the continuous optimization problem in Eq. (3) using minimization oracles of individual discrete functions SFM$\mathbf{D}_i$ in Eq. (5). As a first step, we propose a modified divide-and-conquer algorithm to solve the continuous optimization problem SFM$\mathbf{C}_i$, for each $i \in [r]$ in Section 3.1. In Section 3.2, we use the optimization problems SFM$\mathbf{C}_i$ as black boxes to solve the continuous optimization problem in Eq. (3). The overview is provided in Algorithm 1.

---
**Algorithm 1** From SFM$\mathbf{D}_i$ to SFM$\mathbf{C}$
---
1: **Input** Discrete function minimization oracle for $F_i : 2^V \to \mathbb{R}$ and $\varepsilon \in \mathbb{R}_+$.
2: **output** Optimal primal/dual solutions for Eq. (3) and Eq. (9) respectively $(w^*, s^*)$
3: **for all** $i \in [r]$ **do**
4:      Optimize SFM$\mathbf{C}$ using SFM$\mathbf{C}_i$ by applying algorithms in Section 3.3 and Section 3.4. This requires optimal primal-dual solutions of SFM$\mathbf{C}_i$, $(w_i^*, s_i^*)$ that may be obtained using Algorithm 2 in Section 3.1 assuming oracles SFM$\mathbf{D}_i$.
5: **end for**
6: $(w^*(U), s^*(U)) = (w_U^*, s_U^*)$
---

## 3.1 Single submodular function

For brevity, we drop the subscript $i$ and consider the following primal optimization problem. Algorithm 2 below is an extension of the classical divide-and-conquer algorithm from [29]. Note that it requires access to dual certificates for the SFM problems.

---
**Algorithm 2** From SFM$\mathbf{D}_i$ to SFM$\mathbf{C}_i$
---
1: **Input** Discrete function minimization oracle for $F : 2^V \to \mathbb{R}$ and $\varepsilon \in \mathbb{R}_+$.
2: **output** Optimal primal/dual solutions for Eq. (3) and Eq. (9) respectively $(w^*, s^*)$
3: $A_+ = \operatorname{argmin}_{A \subset V} F(A) + \varepsilon|A|$ with a dual certificate $s_+ \in B(F)$.
4: $A_- = \operatorname{argmin}_{A \subset V} F(A) - \varepsilon|A|$ with a dual certificate $s_- \in B(F)$ (we must have $A_+ \subseteq A_-$)
5: $w^*(A_+) = -\varepsilon, s^*(A_+) = s_+, w^*(V \setminus A_-) = \varepsilon, s^*(V \setminus A_-) = s_-$
6: $U := A_- \setminus A_+$ and a discrete function $G : 2^U$ s.t. $G(B) = F(A_+ \cup B) - F(A_+)$ with Lovász extension $g : \{0,1\}^{|U|} \to \mathbb{R}$.
7: Solve for optimal solutions of $\min_{w \in \mathbb{R}^{|U|}} g(w) + \frac{1}{2}w^2$ and its dual using divide-and-conquer algorithm [16] to obtain $(w_U^*, s_U^*)$.
8: $(w^*(U), s^*(U)) = (w_U^*, s_U^*)$
---

**Proposition 1** *Algorithm 2 gives an optimal primal-dual pair for the optimization problem in Eq. (3) and Eq. (9) respectively.*

See the proof in the supplementary material (Section A). Note that the number of steps is at most the number of different values that $w$ may take (the solution $w$ is known to have many equal components [30]). In the worst case, this is still $n$, but in practice many components are equal to $-\varepsilon$ or $\varepsilon$, thus reducing the number of SFM calls (for $\varepsilon$ very close to zero, only two calls are necessary). In Section 5, we show empirically that this is the case, the number of SFM calls decreases significantly when $\varepsilon$ tends to zero.

### 3.2 Sum of submodular functions

In this section, we consider the optimization problem in Eq. (3) with the function $\psi$ from Eq. (7). The primal optimization problem is given by

$$\min_{w\in[-\varepsilon,\varepsilon]^n} \sum_{i=1}^{r} f_i(w) + \frac{1}{2}\|w\|_2^2. \tag{11}$$

In order to derive a dual problem with the appropriate structure, we consider the functions $g_i$ defined as follows: $g_i(w) = f_i(w)$ if $|w| \leqslant \varepsilon$, and $+\infty$ otherwise, with the Fenchel conjugate

$$g_i^*(s_i) = \sup_{w\in[-\varepsilon,\varepsilon]^n} w^\top s_i - f_i(w) = \inf_{t_i\in B(F_i)} \sup_{w\in[-\varepsilon,\varepsilon]^n} w^\top(s_i - t_i) = \varepsilon \inf_{t_i\in B(F_i)} \|s_i - t_i\|_1.$$

Therefore, we can derive the following dual:

$$\begin{aligned}
\min_{w\in[-\varepsilon,\varepsilon]^n} \sum_{i=1}^{r} f_i(w) + \frac{1}{2}\|w\|_2^2 &= \min_{w\in\mathbb{R}^n} \sum_{i=1}^{r} g_i(w) + \frac{1}{2}\|w\|_2^2 \\
&= \min_{w\in\mathbb{R}^n} \sum_{i=1}^{r} \max_{s_i\in\mathbb{R}^n} \left\{ w^\top s_i - g_i^*(s_i) \right\} + \frac{1}{2}\|w\|_2^2 \\
&= \max_{(s_1,\ldots,s_r)\in\mathbb{R}^{n\times r}} -\sum_{i=1}^{r} g_i^*(s_i) - \frac{1}{2}\Big\|\sum_{i=1}^{r} s_i\Big\|_2^2. \tag{12}
\end{aligned}$$

We are now faced with the similar optimization problem than previous work [15], where the primal problems is equivalent to computing the proximity operator of the sum of functions $g_1 + g_2$. The main difference is that when $\varepsilon$ is infinite (i.e., with no constraints), then the dual functions $g_i^*$ are indicator functions of the base polytopes $B(F_i)$, and the dual problem in Eq. (12) can be seen as finding the distance between two polytopes.

This is not the case for our constrained functions. This limits the choice of algorithms. In this paper, we consider block-coordinate ascent (which was already considered in [15], leading to alternate projection algorithms), and a novel recent accelerated coordinate descent algorithm [27]. We could also consider (accelerated) proximal gradient descent on the dual problem in Eq. (12), but it was shown empirically to be worse than alternating reflections [15] (which we compare to in experiments, but which we cannot readily extend without adding a new hyperparameter).

### 3.3 Optimization algorithms for all $r$

All of our algorithms will rely on the computing the proximity operator of the functions $g_i^*$, which we now consider.

**Proximity operator.** The key component we will need is the so-called proximal operator of $g_i^*$, that is being able to compute efficiently, for a certain $\eta$,

$$\min_{s_i\in\mathbb{R}^n} g_i^*(s_i) + \frac{1}{2\eta}\|s_i - t_i\|_2^2.$$

Using the classical Moreau identity [31], this is equivalent to solving

$$\begin{aligned}
\min_{s_i\in\mathbb{R}^n} g_i^*(s_i) + \frac{1}{2\eta}\|s_i - t_i\|_2^2 &= \min_{s_i\in\mathbb{R}^n} \max_{w_i\in\mathbb{R}^n} w_i^\top s_i - g_i(w_i) + \frac{1}{2\eta}\|s_i\|_2^2 + \frac{1}{2\eta}\|t_i\|_2^2 - \frac{1}{\eta} s_i^\top t_i \\
&= \max_{w_i\in\mathbb{R}^n} -g_i(w_i) - \frac{\eta}{2}\|w_i - \frac{1}{\eta}t_i\|_2^2 + \frac{1}{2\eta}\|t_i\|_2^2 \\
&= \max_{w_i\in\mathbb{R}^n} -g_i(w_i) + w_i^\top t_i - \frac{\eta}{2}\|w_i\|_2^2.
\end{aligned}$$

This is exactly the oracle $\mathrm{SFM}\mathbf{C}_i$, for which we presented in Section 3.1 an algorithm using only the discrete oracles $\mathrm{SFM}\mathbf{D}_i$.

**Block coordinate ascent.** We consider the following iteration

$$\forall i \in [r], \ s_i^{\mathrm{new}} = \underset{s_i^{\mathrm{new}} \in \mathbb{R}^n}{\mathrm{argmin}} \ g_i^*(s_i^{\mathrm{new}}) + \tfrac{1}{2} \big\| \textstyle\sum_{j=1}^i s_j^{\mathrm{new}} + \sum_{j=i+1}^r s_j \big\|_2^2,$$

which is exactly block-coordinate ascent on the dual problem in Eq. (12). Since the non-smooth function $\sum_{i=1}^r g_i^*(s_i)$ is separable, it is globally convergent, with a convergence rate at least equal to $O(1/t)$, where $t$ is the number of iterations (see, e.g., Theorem-6.7 of [32]).

### 3.4 Acceleration for the special case $r = 2$

When there are only two functions, following [27], the problem in Eq. (12) can be written as:

$$\max_{s_1 \in \mathbb{R}^n} -g_1^*(s_1) - h_1(s_1), \tag{13}$$

with

$$
\begin{aligned}
h_1(s_1) &= \inf_{s_2 \in \mathbb{R}^n} g_2^*(s_2) + \tfrac{1}{2}\|s_1 + s_2\|^2 = \sup_{w_2 \in \mathbb{R}^n} \inf_{s_2 \in \mathbb{R}^n} w_2^\top s_2 - g_2(w_2) + \tfrac{1}{2}\|s_1 + s_2\|^2 \\
&= \sup_{w_2 \in \mathbb{R}^n} -g_2(w_2) + \tfrac{1}{2}\|s_1\|_2^2 - \tfrac{1}{2}\|w_2 + s_1\|_2^2, \text{ with } s_2 = w_2 + s_1, \\
&= \sup_{w_2 \in \mathbb{R}^n} -g_2(w_2) - \tfrac{1}{2}\|w_2\|_2^2 - w_2^\top s_1 = \sup_{w_2 \in [-\varepsilon, \varepsilon]^n} -f_2(w_2) - w_2^\top s_1 - \tfrac{1}{2}\|w_2\|_2^2.
\end{aligned}
$$

The function $h_1$ is 1-smooth with gradient equal to $h_1'(s_1) = s_1 + s_2^*(s_1)$. Applying proximal gradient to the problem of maximizing $\max_{s_1 \in \mathbb{R}^n} -g_1^*(s_1) - h_1(s_1)$ leads to the iteration

$$
\begin{aligned}
s_2^{\mathrm{new}} &= \underset{s_2^{\mathrm{new}} \in \mathbb{R}^n}{\mathrm{argmin}} \ g_2^*(s_2^{\mathrm{new}}) + \tfrac{1}{2}\|s_1 + s_2^{\mathrm{new}}\|^2 \\
s_1^{\mathrm{new}} &= \underset{s_1^{\mathrm{new}} \in \mathbb{R}^n}{\mathrm{argmin}} \ g_1^*(s_1^{\mathrm{new}}) + \tfrac{1}{2}\|s_1^{\mathrm{new}} - s_1\|_2^2 + h_1'(s_1)^\top(s_1^{\mathrm{new}} - s_1) \\
&= \underset{s_1^{\mathrm{new}} \in \mathbb{R}^n}{\mathrm{argmin}} \ g_1^*(s_1^{\mathrm{new}}) + \tfrac{1}{2}\|s_1^{\mathrm{new}} - s_1\|_2^2 + (s_1 + s_2^{\mathrm{new}})^\top(s_1^{\mathrm{new}} - s_1) \\
&= \underset{s_1^{\mathrm{new}} \in \mathbb{R}^n}{\mathrm{argmin}} \ g_1^*(s_1^{\mathrm{new}}) + \tfrac{1}{2}\|s_1^{\mathrm{new}} + s_2^{\mathrm{new}}\|_2^2,
\end{aligned}
$$

which is *exactly* block coordinate descent. Each of these steps are exactly using the same oracle as before. We can now accelerate it using FISTA [33] with the step size from the smoothness constant which is equal to 1. Starting from a pair of iterate $(s_1, t_1)$, this leads to the iteration:

$$
\begin{aligned}
s_2^{\mathrm{new}} &= \underset{s_2^{\mathrm{new}} \in \mathbb{R}^n}{\mathrm{argmin}} \ g_2^*(s_2^{\mathrm{new}}) + \tfrac{1}{2}\|t_1 + s_2^{\mathrm{new}}\|^2 \\
s_1^{\mathrm{new}} &= \underset{s_1^{\mathrm{new}} \in \mathbb{R}^n}{\mathrm{argmin}} \ g_1^*(s_1^{\mathrm{new}}) + \tfrac{1}{2}\|s_1^{\mathrm{new}} + s_2^{\mathrm{new}}\|_2^2 \\
t_1^{\mathrm{new}} &= s_1^{\mathrm{new}} + \beta(s_1^{\mathrm{new}} - s_1),
\end{aligned}
$$

with $\beta = (t-1)/(t+2)$ at iteration $t$. This algorithm converges in $O(1/t^2)$.

This acceleration can also be used for the case $r > 2$ by using the product space trick (see, e.g., [15, Section 3.2]). However, this requires a correction in the product space that leads to inefficiencies of the algorithm in practice. See [34] for more details.

## 4 Theoretical Analysis

In this section, we provide a convergence analysis for the methods above. For simplicity of results, we consider the following primal-dual formulation (where both primal and dual variables live in bounded sets):

$$
\begin{aligned}
\min_{w \in [-\varepsilon, \varepsilon]^n} \sum_{i=1}^r f_i(w) + \frac{1}{2}\|w\|_2^2 &= \min_{w \in \mathbb{R}^n} \sum_{i=1}^r \max_{t_i \in B(F_i)} w^\top t_i + \sum_{j=1}^n \psi(w_j) \\
&= \max_{(t_1, \dots, t_r) \in B(F_1) \times \cdots \times B(F_r)} -\sum_{j=1}^n \psi^*\Big(\sum_{i=1}^r t_{ij}\Big). \tag{14}
\end{aligned}
$$

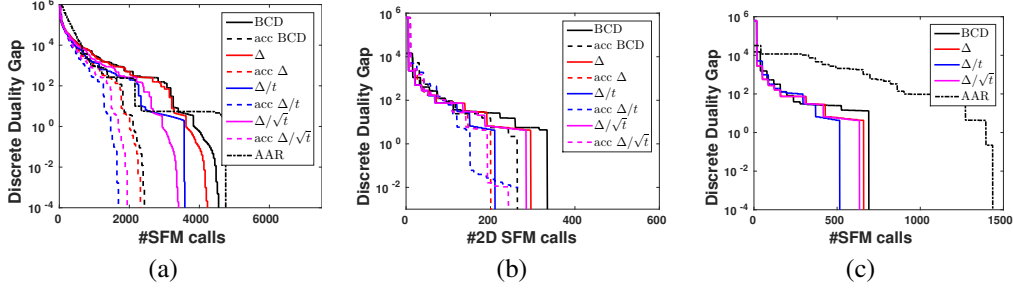

Figure 1: Comparison to state-of-the-art algorithms for 2D and 3D SFM.

We assume that we have a pair $(w, t_1, \ldots, t_r)$ of approximate primal-dual solutions for Eq. (14), with a duality gap $\eta_\mathrm{C}$. This leads to a pair $(w, u)$ of primal-dual approximate solutions for

$$\min_{w \in [-\varepsilon, \varepsilon]^n} f(w) + \frac{1}{2}\|w\|_2^2 = \max_{u \in B(F)} - \sum_{j=1}^n \psi^*(u_j), \qquad (15)$$

for which we can get an approximate subset of $V$ (see proof in supplementary material, Section B):

**Proposition 2** *Given a feasible primal candidate $w$ for Eq. (15) with suboptimality $\eta_\mathrm{C}$, one of the suplevel sets $\{w \geqslant \alpha\}$ of $w$ is an $\eta_\mathrm{D}$-optimal minimizer of $F$, with $\eta_\mathrm{D} = \frac{\eta_\mathrm{C}}{4\varepsilon} + \sqrt{\frac{\eta_\mathrm{C} n}{2}}$.*

Since our dual problems are all $O(1)$-smooth (using the traditional definitions of smoothness [35]), their guarantees will always be of the form $\eta_\mathrm{C} = \frac{\Delta^2}{t^\alpha}$ where $\Delta$ is a notion of diameter of the base polytopes and $\alpha = 2$ for accelerated algorithms and $\alpha = 1$ for plain algorithms. The overall discrete gap is thus up to constant terms

$$\eta_\mathrm{D} = \frac{\Delta \sqrt{n}}{t^{\alpha/2}} + \frac{\Delta^2}{\varepsilon t^\alpha}.$$

We see clearly that the final bound on the (discrete) gap is decreasing with $\varepsilon$. This suggest to use $\varepsilon$ proportional to $\frac{\Delta}{\sqrt{n t^\alpha}}$ to take it as small as possible while only losing a factor of 2 in the convergence bound.

**Guarantees for FISTA applied to the dual of Eq. (14).**   The function $\psi^*$ is $O(1)$-smooth, and the objective in Eq. (14) is $r$-smooth. Each $B(F_i)$ has a square diameter less than $\Delta_i^2 = \sum_{j=1}^n \left[ F_i(\{j\}) + F_i(V \backslash \{j\}) - F_i(V) \right]^2$. Thus, in the result above, we have $\Delta^2 = r \sum_{i=1}^r \Delta_i^2$ and $\alpha = 2$. Owing to [36, Cor. 2(b)], these guarantees extend to the corresponding primal iterate $w$.

**Guarantees for primal-dual algorithms applied to Eq. (14).**   We consider the primal-dual formulation

$$\min_{w \in [-\varepsilon, \varepsilon]^n} \max_{(t_1, \ldots, t_r) \in B(F_1) \times \cdots \times B(F_r)} w^\top \left( \sum_{i=1}^r t_i \right) + \sum_{j=1}^n \psi(w_j).$$

The primal set has squared diameter $n\varepsilon^2$; the dual set has squared diameter less than $\sum_{i=1}^r \Delta_i^2$, the bilinear function has a largest singular value equal to $\sqrt{r}$. Thus, from [37], we get a guarantee from a primal-dual algorithm, of the form $\Delta^2 = r \sum_{i=1}^r \Delta_i^2 + \varepsilon \sqrt{nr} \sqrt{\sum_{i=1}^r \Delta_i^2}$. We thus get overall a guarantee of the same form as above, with the same dependency in $\varepsilon$.

## 5   Experiments

In this section, we consider the minimization of cut functions [3] that are an important examples of submodular functions. In our experiments, we consider the problem of minimizing cuts on 2D images and 3D volumetric surfaces for segmentation. We consider a two-dimensional image of size $n = 2400 \times 2400 = 5.8 \times 10^6$ pixels, and a 3D volumetric surface of size $n = 102 \times 100 \times 79 = 8.1 \times 10^5$ voxels. The SFM oracles are obtained by using max-flow codes, which is the dual of the min-cut problem. We compare our results to the standard block coordinate descent (BCD) [15] and

averaged alternating reflections algorithm (`AAR`) [15], which are using full total variation oracles (which we solve using the usual divide-and-conquer algorithm that is only using SFM calls).

In our approach, we have a parameter $\varepsilon$ dependent on $\frac{\Delta}{\sqrt{n}t^{\alpha}}$, where $\Delta$ is the notion of diameter of the base polytope, $n$ is the number of elements in the ground set and $t$ is the number of iterations. In our experiments, we choose $\varepsilon$ proportional to $\Delta$, $\Delta/t$ and $\Delta/\sqrt{t}$ and respectively represent them by the same terms in Figure 1. For the case of the sum of two functions, the block coordinate descent can be accelerated [27] as shown in Section 3.4. We refer to their accelerated versions as `acc BCD`, `acc` $\Delta$, `acc` $\Delta/t$ and `acc` $\Delta/\sqrt{t}$ respectively. Therefore, `BCD`, `acc BCD`, `AAR` use quadratic $\psi$ and the rest use $\psi$ as defined in Eq. (7).

Figure 1 shows the performance of various algorithms on different problems which we detail below. The horizontal axes represents the number of discrete minimization oracles, i.e., SFM$\mathbf{D}_i$ required to solve the SFM and the vertical axes represents the discrete duality gap given by

$$\text{gap}(A, s) = F(A) - s_-(V),$$

where $A \subset V$, $s \in B(F)$ are the discrete primal-dual pairs and $s_-(V) = \sum_{i=1}^n \min(s_i, 0)$. We consider three experiments that may be broadly classified into sum of two functions and sum of three functions.

**Sum of two functions ($r = 2$).** In this case, we consider minimization of the submodular function that can be written as sum of two submodular functions, i.e., $F = F_1 + F_2$. We consider the problem of mininiming graph cuts on 2D grid that can be written as the sum of horizontal and vertical chain graphs in Figure 1-(a). In this case, the SFM$\mathbf{D}_i$ orcale represents the min-cut on a chain graph while the SFM problem represents min-cut on a 2D grid. We can observe that the constrained total variation formulation reduces the number of min-cut/ max-flow calls when compared to full total variation. Here, we explicitly calculate the diameter of the base polytope $\Delta$ for choosing $\varepsilon$.

Figure 2-(a) shows the total number of SFM$\mathbf{D}_i$ oracle calls required to solve the SFM problem for different values of $\varepsilon$. Figure 2-(b) shows the total number of constrained TV SFM$\mathbf{C}_i$ calls required to solve the SFM problem. Figure 2-(c) shows the average number of SFM$\mathbf{D}_i$ oracle calls required to solve a single SFM$\mathbf{C}_i$ problem. The algorithms considered in these graphs are BCD and accelerated BCD algorithms using constrained total variation represented by $\varepsilon$ and `acc` $\varepsilon$ respectively. We clearly see the trade-off for the choice of $\varepsilon$: the number of SFM calls per TV calls increases with $\varepsilon$, while the number of TV calls decreases, leading to intermediate values of $\varepsilon$ which lead to significant gains in the total number of SFM calls in Figure 2-(a).

We consider 3D grid that can be decomposed into 2D frames and chains graphs. In Figure 1-(b), we show the performance of our algorithm compared to other state-of-the-art algorithms for this decomposition. In this case we use two different discrete oracles SFM$\mathbf{D}_i$, i.e., min-cut on a chain and min-cut on a 2D grid to solve SFM, i.e., min-cut on the 3D grid. We show only the number of oracle calls to min-cut on 2D grids for analysis as they are more expensive than min-cuts on chains.

**Sum of three functions ($r = 3$).** In this case, we consider minimization of the submodular function that can be written as sum of three submodular functions, i.e., $F = F_1 + F_2 + F_3$. Min-cut on the 3D grid can also be seen as sum of chain graphs in three directions, thereby using discrete minimization oracles only of the chain graphs. Figure 1-(c) shows the number of calls to 1D min-cut to solve the 3D min-cut problem using various continuous optimization problems and algorithms. Our approach considerably reduces the number of calls to 1D min-cut (SFM$\mathbf{D}_i$) oracles. Figure 2-(d) shows the total number of 1D min-cuts (SFM$\mathbf{D}_i$) to solve 3D SFM for various values of $\varepsilon$. Figure 2-(e) shows the total number of constrained total variation SFM$\mathbf{C}_i$ calls required to solve the SFM problem. Figure 2-(e) shows the average number of SFM$\mathbf{D}_i$ oracle calls required to solve SFM$\mathbf{C}_i$ for this problem. We observe a similar behavior than for $r = 2$, with best values of $\varepsilon$ not being very small or very large.

## 6 Conclusion

In this paper, we have proposed a simple modification of state-of-the-art algorithms for decomposable submodular function minimization. Adding box constraints to the continuous optimization problems allow for significant reduction in the number of individual submodular function minimization calls.

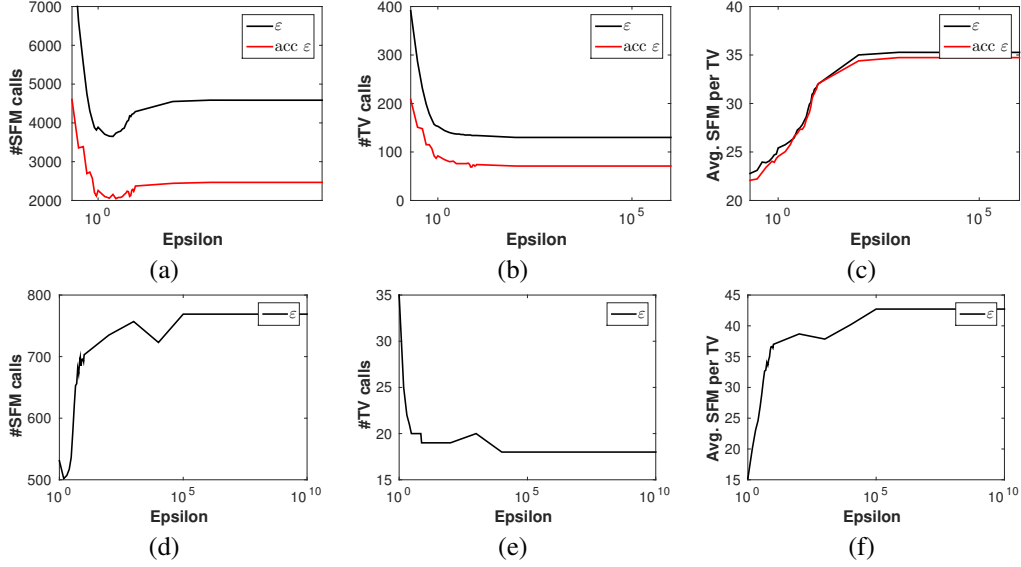

Figure 2: `BCD` and `acc BCD` for 2D function: $F = F_1 + F_2$ and 3D functions: $F = F_1 + F_2 + F_3$.

The application of accelerated block coordinate ascent techniques makes the speed-up stronger. Further speed-ups may be achieved by extended the proposed algorithms to [18, 25]. These techniques are easily parallelizable and it would be interesting to compare to dedicated parallel algorithms for graph cuts [38]. Moreover, these speed-ups could be extended to more general submodular optimization problems [39].

**Acknowledgments.** This research was funded by the Leverhulme Centre for the Future of Intelligence, Cambridge and the Data Science Institute, Imperial College London. We acknowledge support from the European Research Council (SEQUOIA project 724063) and (HOMOVIS project 640156).

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
