[Supplementary Material]

## A Proof of Convergence for Divide-and-Conquer Algorithm

We claim the following:

(a) $A_+ \subset A_-$.

(b) $v \in \mathbb{R}^d$ defined as $v_{A_+} = \varepsilon$, $v_{V\backslash A_-} = -\varepsilon$ and $v_{A_+\backslash A_-} = w_{A_+\backslash A_-}$ is the unique global optimizer of Eq. (3).

(c) $t \in \mathbb{R}^d$ defined so that $t_{A+} = (s_+)_{A_+}$, $t_{V\backslash A_-} = (s_-)_{V\backslash A_-}$ and $t_{A_-\backslash A_+} = s_{A_-\backslash A_+}$, is one of the maximizers of Eq. (9).

The statement (a) is a consequence of the usual results on minimizing $f(w) + \frac{1}{2}\|w\|_2^2$ and its relationship with SFM. We are going to show (b) and (c) by showing that this is a primal/dual pair with equal objectives.

We need to show that $t \in B(F)$. We have for any $A \subset V$, using submodularity twice,

$$
\begin{aligned}
t(A) &= s_+(A\cap A_+) + t(A\cap(A_-\backslash A_+)) + s_-(A\cap(V\backslash A_-)) \\
t(A) &\leqslant F(A\cap A_+) + F(A_+ \cup (A\cap(A_-\backslash A_+))) - F(A_+) + s_-(A\cap(V\backslash A_-)) \\
&\leqslant F(A\cap A_-) + s_-((A\cup A_-)\backslash A_-) \\
&\leqslant F(A\cap A_-) + F(A\cup A_-) - s(A_-) \\
&= F(A\cap A_-) + F(A\cup A_-) - F(A_-) \leqslant F(A).
\end{aligned}
$$

For $A = V$, all inequalities above are equalities, and thus $t \in B(F)$.

We need to show that $w$ is feasible, i.e., that $w_{A_+\backslash A_-}$ has components in $[-\varepsilon, \varepsilon]$. This is a consequence of classical results for minimizing $f(w) + \frac{1}{2}\|w\|_2^2$ [4, Prop. 8.1]

We can then compute the Lovász extension values exactly and we then have a primal value equal to:

$$
\begin{aligned}
&f(v) + \sum_{i=1}^{n} \psi(v_i) \\
=\ &f(v) + \frac{1}{2}\|v\|^2 \\
=\ &\varepsilon F(A_+) + f_{A_-}^{A_+}(w_{A_+\backslash A_-}) + \varepsilon F(V) - \varepsilon F(A_-) + \frac{1}{2}\|w_{A_+\backslash A_-}\|^2 + \frac{\varepsilon^2}{2}|A_+| + \frac{\varepsilon^2}{2}|V\backslash A_-|.
\end{aligned}
$$

The dual value is equal to

$$
\begin{aligned}
&-\sum_{i=1}^{n} \psi^*(-t_i) \\
=\ &-\sum_{i\in A_+} \psi^*(-t_i) - \sum_{i\in A_-\backslash A_+} \psi^*(-t_i) - \sum_{i\in V\backslash A_-} \psi^*(-t_i) \\
=\ &-\varepsilon\sum_{i\in A_+}\left(|(s_+)_i| - \frac{\varepsilon}{2}\right) + f_{A_-}^{A_+}(w_{A_+\backslash A_-}) + \frac{1}{2}\|w_{A_+\backslash A_-}\|^2 - \varepsilon\sum_{i\in V\backslash A_-}\left(|(s_-)_i| - \frac{\varepsilon}{2}\right) \\
=\ &\varepsilon s_+(A_+) + \frac{\varepsilon^2}{2}|A_+| + f_{A_-}^{A_+}(w_{A_+\backslash A_-}) + \frac{1}{2}\|w_{A_+\backslash A_-}\|^2 + \varepsilon s_-(V\backslash A_-) + \frac{\varepsilon^2}{2}|V\backslash A_-|,
\end{aligned}
$$

which is thus equal to the primal value, hence optimality. Here we have used the fact that $s_+(A_+) + s_-(V\backslash A_-) = F(V) + F(A_+) - F(A_-)$. Indeed, $s_+$ is the dual certificate for a SFM problem, and has to satisfy $s_+(A+) = F(A_+)$ [4, Prop. 10.3]. Similarly, $s_-(A_-) = F(A_-)$, which leads to $s_+(A_+) + s_-(V\backslash A_-) = s_+(A_+) + s_-(V) - s_-(A_-) = F(A_+) + F(V) - F(A_-)$.

Note that in the algorithm, there are some free choices for $s_+$ and $s_-$, and that we can take all of them as subvector of the dual to the minimization of $f(w) + \frac{\varepsilon}{2}\|w\|_2^2$, but this is not the only choice.

## B Proof of Prop. 2

We follow the proof of [4, Prop. 10.5], which corresponds to the case $\varepsilon = +\infty$.

From a feasible primal candidate, we can always build a dual candidate $s$ (e.g., by taking any dual maximizer). If we assume that for all $\alpha \in [-c, c]$, for $c \in [0, \varepsilon]$ we have $(F + \psi'(\alpha))(\{w \geqslant \alpha\}) - (s + \psi'(\alpha))_-(V) > \eta_C/(2c)$, then we obtain that

$$\eta_C \geqslant \int_{-c}^{c} (F + \psi'(\alpha))(\{w \geqslant \alpha\}) - (s + \psi'(\alpha))_-(V) d\alpha > \eta_C,$$

which is a contradiction. Thus, we must at least one $\alpha \in [-c, c]$ such that $(F + \psi'(\alpha))(\{w \geqslant \alpha\}) - (s + \psi'(\alpha))_-(V) \leqslant \eta_C/(2c)$. This implies that

$$F(\{w \geqslant \alpha\}) - s_-(V) \leqslant \eta_C/(2c) + cn.$$

This means that at least one level set of $w$ has a certified gap less than

$$
\begin{aligned}
\eta_D &= \inf_{c \in [0, \varepsilon]} \eta_C/(2c) + cn \\
&= \inf_{c \in [0,1]} \eta_C/(2\varepsilon c) + cn\varepsilon \\
&= (2n\varepsilon) \times \inf_{c \in [0,1]} \frac{1}{2}(c + \frac{1}{c}\frac{\eta_C}{4n\varepsilon^2}) \\
&\leqslant (2n\varepsilon) \times \left(\sqrt{\frac{\eta_C}{4n\varepsilon^2}} + \frac{1}{2}\frac{\eta_C}{4n\varepsilon^2}\right) = \sqrt{\eta_C n/2} + \eta_C/(4\varepsilon)
\end{aligned}
$$

using the identity $\inf_{c \in [0,1]} \frac{1}{2}(c + \frac{a}{c}) \leqslant \sqrt{a} + a/2$. If $a <= 1$, take $c = \sqrt{a}$ and the inf is less than $\sqrt{a}$ and thus less than $\sqrt{a} + a/2$. If $a >= 1$, take $c = 1$ and the inf is less than $1/2 + a/2$, which is less than $\sqrt{a} + a/2$ because $1/2 < \sqrt{a}$