[Reviews · NeurIPS 2019]

Reviewer 1



After reading the rebuttal and other reviews, my score remains the same. ======== Overall the submission presents a novel, simple-yet-non-obvious change to existing approaches that can yield substantial improvements. The change is well-motivated and accompanied by appropriate supporting theory, which is correct to my knowledge. The experiments are appropriate: the authors show improved performance, and also study the sensitivity of performance to the choice of the constraint \epsilon (showing of course that there is a tradeoff to optimize). On the whole it is a solid paper and I recommend acceptance. Some small comments: - There is a minor typo in the displayed equation after line 376 around the $s$. - It would be good to explain the identity in line 378. - Please be specific about what part of [31] is being referenced in line 166.

Reviewer 2



Originality: It is a neat idea to introduce a different proximal term of the continuous variable w, which may help to improve the efficiency of the algorithm. This idea in a high level works in a similar way as that of screening elements and keeping a small active set, e.g. the one adopted in reference [25]. Other analysis and discussion seem to be a natural generalization of previous works, e.g., [15]. Therefore, I think the originality is around the borderline. Quality: The submission is technically sound. Regarding the novel term on w, the authors provide a relatively complete discussion on how to handle it, although some techniques are overlapping with those adopted in the previous work [15]. Clarity: The clarity of this work is not very good. As a reader that is very familiar with the important references, I have to read through Section 3 several times to get a clear outline of how the overall algorithm works and where the potential improvement comes from. The problem is caused by the fact that there are too many formulations including discrete, continuous ones and different algorithms for different setting proposed by the authors. I think it would be difficult for a non-expert/practitioner to follow the idea clearly. Significance: I think the significance of the work is not extraordinary but okay. The idea is good. Although it is a small idea and kind of overlapping other ideas at a high level, it is good to see in this particular setting and indeed improves the algorithms. ---After reading the rebuttal --- I agree with other reviewers that the contribution of this work is solid although the change is slight. However, there is one additional minor comment: Since the RBCD methods proposed in [17,18] hold theoretical characterization of the number of iterations needed while the cyclic BCD used here does not hold the similar result (also questioned by Reviewer 4), why not directly use RBCD? A table to list the whole algorithm can be rather helpful. Please provide the evaluation over the cases with more complexity potential functions as promised.

Reviewer 3



Comments and Questions: Originality: The paper presents a constrained version of the \psi function for minimizing the sum of submodular set functions. As far as I know, I did not see this technique before. Quality: Experiments support the fast convergence of proposed algorithms. For the theoretical results, can you get a faster rate than previous algorithms, say, [15] and [17]? Meanwhile, is it possible to give oracle complexity in terms of either discrete minimization or function value evaluations? This would be a good way to theoretical justify the advantage of your algorithm. Clarity: Overall, the paper is well-presented. Significance: Submodular minimization is well applied in segmentation and constraint satisfaction, this technique would be a good way to accelerate the SFM process. - The choice of \epsilon depends on diameter of the base polytope D. Is there an efficient method to estimate it? - From the experiments you observe a tradeoff of choosing \epsilon, do you have some heuristic method to choose it. - Do you see a hope of applying this technique to minimizing submodular functions over continuous domains? Minor comments: - line 68: loop to the solve -> loop to solve -line 196: less than less -> less than

[Author Response · NeurIPS 2019]

We thank the reviewers for the feedback. We will ensure that all the minor issues are corrected in the final version of the
draft. Please find the responses for other comments/queries below.

R2  • Explain the identitiy $\inf_{c\in[0,1]} \frac{1}{2}(c + \frac{a}{c}) \leqslant \sqrt{a} + a/2$.
– if $a <= 1$, take $c = \sqrt{a}$ and the $\inf$ is less than $\sqrt{a}$ and thus less than $\sqrt{a} + a/2$;
– if $a >= 1$, take $c = 1$ and the $\inf$ is less than $1/2 + a/2$, which is less than $\sqrt{a} + a/2$ because $1/2 < \sqrt{a}$
• Please be specific about what part of [31] is being referenced in line 166. It is Theorem-6.7 of page 336 of [31],
for the randomized version. We will be more precise in the final version.

R3  • Provide an outline table to clearly illustrate how the overall algorithm looks like: To solve the original problem,
does one have to solve SFMCi iteratively? To solve SFMCi, does one have need algorithm 1 and solving SFMDi
several times? We mentioned the gist of the algorithms in line 66-69.
"In Section 3.1, we show that the continuous optimization problem $\text{SFM}\mathbf{C}_i$ can be optimized using discrete
oracles $\text{SFM}\mathbf{D}_i$ using a modified divide-and-conquer algorithm [15]. In Section 3.2, we use various optimization
algorithms that use $\text{SFM}\mathbf{C}_i$ as the inner loop to the solve the continuous optimization problem in Eq.(3)
consequently solving the SFM problem in Eq.(1)."
We would be glad to reiterate this before section 3, in particular in more "algorithmic" form, if this improves
the clarity of the paper.
• The block coordinate descent method (BCD) discussed in Section 3.3 is cyclic BCD. The BCD in section 3.4 is
still cyclic, while it is a reduced version for the case r=2. The authors would like to check the RCD method and
its accelerated version in references [17,18]. Although each step of descent therein is to compute projection
to base polytope which is different from this work due to the new proximal term on w, the authors would
like to give a discussion on whether that method may be used here or even provide additional experimental
evaluations on RCD. This may easily be extended to Randomized coordinate descent method [17,18]. However,
the possibility of extending it to the accelerated version of the randomized coordinate descent [17] is non-trivial
and need some more time.
• The experiments only consider chain-graphs as decomposed parts. In reference [15, 18], the DSFM problems
may include much more complex potentials defined over superpixels. Please provide the results with such
settings. We ll try to add results for a problem where one of the summand functions as a concave function over
super pixels instead of just chain or frame functions in the final version of the draft.
• Other than just the number of iterations, it would be better to have CPU times of different algorithms for
comparison as well.
The CPU times are proportional to the number of SFM calls. We would be glad to add the CPU times in the
final version of the draft. The CPU times for 3D SFM decomposed into 3 functions is BCD - 17.40 sec, $\Delta$
- 15.60 sec, $\Delta/t$ - 10.09 sec, $\Delta/\sqrt{t}$ - 12.27 sec respectively. Note that the timings might change with better
hardware with high parallelism. These timings are on a 4-core laptop processor(Intel(R) Core(TM) i7-7600U
CPU @ 2.80GHz).
• A recent work "Revisiting Decomposable Submodular Function Minimization with Incidence Relations" by
Li et al. in NeurIPS 2018 also considered using different terms of w to accelerate the algorithm for DSFM
problems. Thank you for the reference, we will add it. This seems like an interesting direction for future work
of being able to further speed up the process and come up with better algorithms

R4  • For the theoretical results, can you get a faster rate than previous algorithms, say, [15] and [17]? Meanwhile, is
it possible to give oracle complexity in terms of either discrete minimization or function value evaluations? This
would be a good way to theoretical justify the advantage of your algorithm. We do not yet have a conclusive
proof that can show faster convergence rate than the previous algorithms [15,17]. However, the number of
discrete minimization oracle calls is $O(|U|)$, where $U = A_- \setminus A_+$ in Algorithm 1 which is empirically much
lesser than $O(|V|)$ for smaller $\varepsilon$, but for which we currently do not have a bound.
• The choice of $\varepsilon$ depends on diameter of the base polytope $\Delta$. Is there an efficient method to estimate it? From
the experiments you observe a tradeoff of choosing $\varepsilon$, do you have some heuristic method to choose it. We
suggest to use $\varepsilon$ proportional to $\frac{\Delta}{\sqrt{nt^\alpha}}$ to take it as small as possible while only losing a factor of 2 in the
convergence bound, where $\Delta_i^2 = \sum_{j=1}^n \left[ F_i(\{j\}) + F_i(V\setminus\{j\}) - F_i(V) \right]^2$ thus, in the result above, we have
$\Delta^2 = r \sum_{i=1}^r \Delta_i^2$, where $r$ is the number of summand functions and $\alpha = 2$. Therefore, $\Delta$ may be estimated in
$O(n)$ function evaluations. $\Delta$ may be approximated for cut functions using weights on the edges($E$) of the
graphs using $O(|E|)$ operations.
• Do you see a hope of applying this technique to minimizing submodular functions over continuous domains?
We certainly believe this technique may be used to minimize continuous submodular functions and approximate
submodular functions. This is the future area of research we are looking into.

[Meta-Review · NeurIPS 2019]

All reviewers appreciated the idea of this paper as simple yet effective. In the camera ready, please take into account the reviewer comments, in particular reviewer #3.